# A Computational Model for Tail Undulation and Fluid Transport in the Giant Larvacean

Alexander P. Hoover [1,*] , Joost Daniels [2] , Janna C. Nawroth [3,*,†] and Kakani Katija [2,*,†]

1   Department of Mathematics, University of Akron, Akron, OH 44325, USA
2   Research and Development, Monterey Bay Aquarium Research Institute, Moss Landing, CA 95039, USA; joost@mbari.org
3   Keck School of Medicine, University of Southern California, Los Angeles, CA 90033, USA
*   Correspondence: ahoover1@uakron.edu (A.P.H.); jnawroth@gmail.com (J.C.N.); kakani@mbari.org (K.K.)
†   These authors contributed equally to this work.

**Abstract:** Flexible propulsors are ubiquitous in aquatic and flying organisms and are of great interest for bioinspired engineering. However, many animal models, especially those found in the deep sea, remain inaccessible to direct observation in the laboratory. We address this challenge by conducting an integrative study of the giant larvacean, an invertebrate swimmer and "fluid pump" of the mesopelagic zone. We demonstrate a workflow involving deep sea robots, advanced imaging tools, and numerical modeling to assess the kinematics and resulting fluid transport of the larvacean's beating tail. A computational model of the tail was developed to simulate the local fluid environment and the tail kinematics using embedded passive (elastic) and active (muscular) material properties. The model examines how varying the extent of muscular activation affects the resulting kinematics and fluid transport rates. We find that muscle activation in two-thirds of the tail's length, which corresponds to the observed kinematics in giant larvaceans, generates a greater average downstream flow speed than other designs with the same power input. Our results suggest that the active and passive material properties of the larvacean tail are tuned to produce efficient fluid transport for swimming and feeding, as well as provide new insight into the role of flexibility in biological propulsors.

**Keywords:** fluid-structure interaction; larvacean; biological propulsion; fluid pump

## 1. Introduction

Natural systems have evolved to exhibit many different mechanisms for fluid transport that aid in every step of their survival. This is particularly true in the areas of locomotion and feeding, where animals are in an evolutionary arms race to outperform their predators and prey. For this reason, there is significant interest in studies of biological fluid transport to leverage nature's know-how for robotics and bioinspired design [1]. Animal-fluid interaction studies have largely focused on animals that can be maintained in a laboratory environment. Recent developments that improve access to other biological models—particularly in underwater environments—are opening up new lines of inquiry [2–6]. The recognition that swimming animals create coherent structures in their wakes [7,8] has led to more detailed studies of how particular morphological features, like peduncles in whales [9] and denticles in sharks [10], can play a role in drag reduction and ultimately swimming performance. Many of these findings, including the discovery of resonance matching between body kinematics and fluid dynamics [11,12], which enables completely passive propulsion as demonstrated in dead fish [13], have important implications for the development of engineered systems, such as underwater vehicles.

Early and continuing efforts that link biological swimming performance with fluid-structure interactions have largely focused on marine mammals and fishes. More recent studies involving less common model organisms are also yielding valuable insights into

not only their biomechanics but also their ecology. Studies in jellyfish [14] have revealed that these animals are extremely efficient metazoan swimmers through the combined action of elastic energy storage in body tissues [15], simultaneous contraction of radial musculature [16], and harvesting of energy present in their wake [15]. Ctenophores, another gelatinous zooplankton, propel themselves through water by metachronal movements of ctene rows [17,18], and their hydrodynamically "silent" swimming modes results in more efficient prey capture [19]. Euphausiids (or krill), another metachronal swimmer [20], are known for swimming in large schools, where their position relative to their neighbors has implications for their energy expenditure [21,22]. While these organismal groups are important to study, researchers continually face limitations in understanding these systems due to their inherent morphological and behavioral complexity. For instance, while animals, like jellyfish, are often assumed to be radially symmetric (such that two-dimensional measurement techniques suffice), common behaviors, such as turning, break this symmetry [14,23], and more complex methods combining experimental and numerical techniques are required to understand the fluid-structure interactions [24–26].

Although studying individual organisms can highlight key features, an integrative approach that synthesizes observations across multiple body forms and across taxa can reveal powerful mechanisms that were previously unknown. By looking at jellyfish bell morphology, bell size, and muscle arrangement, the constraints imposed by these structural features on fluid transport function have been shown to predict the different propulsion modes observed across medusae [16]. Looking at the propulsors of swimmers and flyers across taxa has revealed that these animals exhibit a narrow range of bending modes [27]. Regardless of propulsor type, fins or wings consistently demonstrated an inflection point at 2/3 the distance from the base of the structure [27], and it remains unclear what the specific fluid interactions underlying this universal pattern of bending modes for propulsion are. Comparisons of cruising animals across taxa shows that these animals occupy a narrow range of Strouhal numbers (between 0.2 and 0.4) that is tuned for high power [28], and that energetic efficiency is determined by drag on wings and fins [29]. While much of the focus of these efforts and the community at large has been on swimming, animals are involved in a number of different activities (e.g., feeding) that are required for their survival. Are there fluid transport mechanisms involved in feeding or other behaviors that we are missing if we continue to only focus on animal locomotion?

Larvaceans, or appendicularians, form an intriguing model system for investigating animal-fluid interaction because of the integral role their kinematics have in their ability to both swim and feed. Larvaceans are animals with a simple body plan comprised of a tail and a trunk, and they possess specialized cells that secrete mucus to build complex filtration structures [30]. Tail movements play a role not only in free-swimming but also in expanding and pumping fluid through the filtration structures (so-called mucus houses) that they inhabit [31]. Larvacean body lengths range from less than 1 cm up to 10 cm [31,32], thereby spanning a range of Reynolds numbers from 1 to 800 [3,33]. In larvaceans, as well as in larvae of the closely related solitary tunicates, two bands of striated muscle cells flank the "hydrostatic skeleton" of the tail, the so-called notochord. The two muscle bands are known to contract alternatively to bend the tail and result in the beating of the tail [34–37]. Observations of the smallest (genus *Oikopleura* [38]) to largest (genus *Bathochordaeus* [3]) larvaceans reveal species-dependent tail kinematics during in-house pumping [3,38], which is most likely due to variations in the layouts of muscle and neuronal innervation [35], and Reynolds number constraints [3,33,38]. Interestingly, an inflection point is present along the tail of the largest-known larvacean *Bathochordaeus mcnutti*, at the same location (2/3 along the tail's length) [3] as other swimmers and flyers [27]; however, it is not apparent in smaller oikopleuriids [38]. Strouhal numbers of the giant larvaceans range roughly from 0.2 to 0.7 (cf. Reference [3]), and it remains unclear how these kinematics will alter the flow field induced by the feeding animal. Given that the larvacean's beating tail is similar to a classic flapping flexible foil [3,39], investigating these questions is attractive from both an experimental and computational perspective.

Flexible foils have been the subject of a number of computational studies investigating how the foil's passive material properties affect the propulsion performance for a number of different actuation strategies [40–43]. Swimming and pumping both require a transfer of momentum from an organism to the fluid environment, making the interplay between rigid and flexible (or passive and active) material properties a particularly important area of investigation. Detailed numerical models describing lamprey and jellyfish muscle mechanics [44–47] have allowed us to explore these elastohydrodynamic systems where the kinematics are emergent, rather than prescribed, and resulting from the interaction of the passive elastic and active muscle material properties, as well as the surrounding fluid environment.

By using the giant larvacean *B. mcnutti* as a unique model system, we can work to understand the interactions between the tail's structural rigidity and flexibility, muscle actuation, and fluid forces that underlie the tail kinematics. This will elucidate how the ubiquitous 2/3 inflection point seen in numerous flapping propulsors impacts fluid transport and feeding performance in a swimming and pumping animal. As giant larvaceans are found at depths that are unreachable to divers, we will rely on deep sea robots and advanced imaging tools to observe animal kinematics and feeding performance. From these in situ observations, we can numerically simulate a fluid propulsor with passive and muscle actuated regions to understand the effect of material properties and geometry on tail movement and fluid transport. These findings will ultimately inform our understanding of organismal ecology, as well as the mechanistic underpinnings of fluid-structure interactions.

## 2. Materials and Methods

### 2.1. Field Quantification of Larvacean Kinematics

To quantify giant larvacean kinematics and fluid transport performance, we utilized robotic platforms along with a laser-based imaging tool (DeepPIV) that enabled clear observations of tail movements inside the animal's mucus house [3]. DeepPIV consists of a laser housing deployed via a rigid arm that attaches to a remotely operated vehicle (ROV). Within the laser housing is a continuous, 1-W, 671-nm laser (Laserglow Technologies, Toronto, Canada) and line-generating optics (Edmund Optics, Barrington, NJ, USA) that illuminate a sheet of light approximately 1 mm thick in front of the ROV science camera (Mini Zeus II, Insite Pacific Incorporated, San Diego, CA, USA).The laser sheet plane is approximately 50 cm in front of the camera dome, and the laser sheet optics can illuminate an area as large as 20 cm × 20 cm in front of the camera. The ROV science camera records high-definition (1920 pixel × 1080 pixel), progressive format video at 60 frames per second, and the camera housing and optics are specially designed to minimize image distortion. The science camera has a 10× optical zoom, allowing for image fields of view ranging in size from 13 cm × 7 cm to 165 cm × 90 cm while focused on the laser sheet. The videos were recorded on external drives (AJA Video Systems, Grass Valley, CA, USA) using the camera's high-definition multimedia interface output and stored for further data analysis.

Criteria for selecting clips to measure tail kinematics and particle streak length included the following: (i) limited ROV motion, (ii) accurate positioning of the laser sheet with respect to the organism (that is, the pumping tail and trunk needed to be bisected by a 1-mm-thick laser sheet), and (iii) conditions (i) and (ii) were met for at least a single pumping cycle of the tail. These are similar criteria used by other studies conducting in situ fluid motion measurements induced by swimming zooplankton [3,48,49]. The animal was confirmed to be bisected by the laser sheet when the largest cross-sectional area of the trunk, which included the mouth at the animal body's centerline, was shown. Due to its high contrast and imaging capability within mucus structures [5], DeepPIV technology was used for capturing animal kinematics and morphometrics.

### 2.2. Fluid–Structure Interaction Model

To simulate the inflection point behavior we see in the larvacean kinematics, we assume that the tail wave is driven by a wave of active tension traveling from the base

of the tail to the inflection point, which mimics the neuronal innervation of the muscle bands [36,50]. The resulting motion of the entire tail is influenced by the interplay between the active tension and the tail's elastic properties, and the surrounding fluid environment. In this subsection, we discuss the fluid-structure interaction framework, while the material model is further elaborated in the following subsection.

The fluid-structure interaction system is modeled using an immersed boundary (IB) framework. The IB framework employs an Eulerian description of the equations of fluid motion and a Lagrangian frame to describe a deformable immersed boundary or body [51–53]. Here, $\mathbf{X} = (X, Y, Z) \in U$ represents the Lagrangian coordinate system of the immersed structure, where $U$ is denoting the Lagrangian coordinate domain, and $\mathbf{x} = (x, y, z) \in \Omega$ denote physical Cartesian coordinates, with $\Omega$ denoting the physical region of the fluid-structure system. The physical position of the material point $\mathbf{X}$ at time $t$ is denoted with the mapping $\chi(\mathbf{X}, t) \in \Omega$, such that the physical region of the structure at time $t$ is $\chi(U, t) \subset \Omega$.

The IB formulation of the equations of motion is given by:

$$\rho \left( \frac{\partial \mathbf{u}(\mathbf{x}, t)}{\partial t} + \mathbf{u}(\mathbf{x}, t) \cdot \nabla \mathbf{u}(\mathbf{x}, t) \right) = -\nabla p(\mathbf{x}, t) + \mu \nabla^2 \mathbf{u}(\mathbf{x}, t) + \mathbf{f}(\mathbf{x}, t), \tag{1}$$

$$\nabla \cdot \mathbf{u}(\mathbf{x}, t) = 0, \tag{2}$$

$$\mathbf{f}(\mathbf{x}, t) = \int_U \mathbf{F}(\mathbf{X}, t) \, \delta(\mathbf{x} - \chi(\mathbf{X}, t)) \, d\mathbf{X}, \tag{3}$$

$$\int_U \mathbf{F}(\mathbf{X}, t) \cdot \mathbf{V}(\mathbf{X}) \, d\mathbf{X} = -\int_U \mathbb{P}(\mathbf{X}, t) : \nabla_{\mathbf{X}} \mathbf{V}(\mathbf{X}) \, d\mathbf{X} + \int_U \mathbf{G}(\mathbf{X}, t) \cdot \mathbf{V}(\mathbf{X}) d\mathbf{X}, \tag{4}$$

$$\frac{\partial \chi(\mathbf{X}, t)}{\partial t} = \int_\Omega \mathbf{u}(\mathbf{x}, t) \, \delta(\mathbf{x} - \chi(\mathbf{X}, t)) \, d\mathbf{x}, \tag{5}$$

where $\mu$ is the dynamic viscosity of the fluid, $\rho$ is fluid density, $\mathbf{u}(\mathbf{x}, t) = (u_x, u_y, u_z)$ is the Eulerian material velocity at $\mathbf{x}$, $p(\mathbf{x}, t)$ is the Eulerian pressure field, and $\mathbf{G}(\mathbf{X}, t)$ represents an external body force. $\mathbf{F}(\mathbf{X}, t)$ and $\mathbf{f}(\mathbf{x}, t)$ represent the Lagrangian and Eulerian force densities, respectively. $\mathbf{F} = (F_x, F_y, F_z)$ is defined with respect to the first Piola Kirchhoff stress tensor, $\mathbb{P}$, in Equation (4) using a weak formulation, in which $\mathbf{V}(\mathbf{X})$ is an arbitrary Lagrangian test function. The Dirac delta function $\delta(\mathbf{x})$ is the kernel of the integral transforms of Equations (3) and (5). Here, Equation (3) couples the Lagrangian force density to the local Eulerian force density, while Equation (5) enforces the no-slip boundary condition of the structure with respect to the local fluid velocity.

This study uses a hybrid finite difference/finite element version of the immersed boundary (IB/FE) to approximate Equations (1)–(5). The IB/FE method uses a finite difference formulation for the Eulerian equations and a finite element formulation for the Lagrangian structure. More details of the IB/FE method can be found in Reference [54].

### 2.3. Material Model

In this study, the kinematics of the larvacean tail model emerge from the interaction between the assumed material properties of the tail, a tethering force present at the base of the tail, and the resulting local fluid dynamics. The body's material properties are described using the first Piola-Kirchhoff (PK1) stress tensor

$$\mathbb{P} = \mathbb{P}_p + \mathbb{P}_a,$$

where $\mathbb{P}_p$ represents the passive material properties, and $\mathbb{P}_a$ represents the active material properties.

The tail's passive material properties, which represent the tail's elastic properties, are described using a PK1 description of a Neo-Hookean material:

$$\mathbb{P}_p = \eta \mathbb{F} + (\lambda \det(J) - \eta) \mathbb{F}^{-T}, \tag{6}$$

where $\mathbb{F} = \frac{\partial \chi}{\partial X}$ is the deformation gradient of the body, $J$ is the Jacobian of $\mathbb{F}$, $\eta$ is the shear modulus, and $\lambda$ is the bulk modulus. The shear and bulk moduli are defined, respectively, as

$$\eta = \frac{E}{2(1+\nu)},\tag{7}$$

and

$$\lambda = \frac{E\nu}{(1+\nu)(1-2\nu)},\tag{8}$$

where $E$ is the Young's modulus, and $\nu$ is the Poisson ratio.

The larvacean tail is driven by applying a traveling wave of active stress that alternates between the upper and lower vertical half of the tail, mimicking the opposing action of two parallel muscle layers [34,35]. The active material properties describe the muscle stress that drives the tail motion:

$$\mathbb{P}_a = JT\mathbb{F}\mathbf{f_0}\mathbf{f_0}^T,\tag{9}$$

in which $\mathbf{f_0}$ is the (fiber) direction vector of the prescribed tension with respect to the reference configuration, and $T = T(\mathbf{X}, t)$ is the magnitude of tension applied at point $\mathbf{X}$ at time $t$. Here, $\mathbf{f_0} = (1, 0, 0)$ is chosen to model the transverse orientation of the larvacean musculature in the undeformed configuration.

The applied tension is defined as:

$$T(\mathbf{X}, t) = T_{max}\gamma(\mathbf{X}, t) \qquad \gamma(\mathbf{X}, t) = \alpha(X)\beta(\mathbf{X}, t),\tag{10}$$

where $\gamma(\mathbf{X}, t)$ represent the activation strength of the applied tension at point $\mathbf{X}$ at time $t$, $\alpha(X)$ describes the spatial extent of applied tension in the transverse direction, $\beta(\mathbf{X}, t)$, describes the temporal dynamics of the applied tension, and $T_{max}$ is the maximum tension magnitude applied on the tail. Note that $0 \leq \alpha, \beta, \gamma \leq 1$. Here, the spatial parametrization of where an active stress is applied is described using the following:

$$\alpha(X) = \left(1 - \frac{1}{1 + e^{\sigma_a(X-X_T)}}\right)\left(\frac{1}{1 + e^{\sigma_b(X-AL)}}\right),\tag{11}$$

where $X_T$ is the horizontal extent of the region where a tethering force is applied, $L$ is length of the tail, $0 \leq A \leq 1$ is the relative portion of the tail length where muscle tension is applied, and $\sigma_{a,b}$ describe the transition from a region when tension is applied to one where no tension is applied. Note that $\alpha$ depends on $A$ and, given initial body coordinates $\mathbf{X}$, maps the transition from regions where applied tension is absent (such as the tether region and the tail tip) to where it is present.

The temporal dynamics depend on:

$$\beta(\mathbf{X}, t) = \begin{cases} \sin \Theta & \text{if } \sin \Theta > 0 \\ 0 & \text{if } \sin \Theta < 0, \end{cases}\tag{12}$$

where

$$\Theta = \begin{cases} 2\pi(ft - \psi) - \phi & \text{if } \Theta > 0 \\ 0 & \text{if } \Theta < 0, \end{cases}\tag{13}$$

where $\psi = \frac{X}{\lambda}$ for $\lambda$ wavelength of the active tension wave, $f$ is the frequency, and $\phi = 0$ if $Z > 0$ and $\phi = \pi$ if $Z < 0$, so that the top and bottom layer are out of phase with one another.

Additionally, a body force tethers the tip of the larvacean tail to prevent the tail from moving forward. The body force is a stiff tether spring force dependent on the difference between the current and initial spatial coordinates,

$$\mathbf{G}(\mathbf{X}, t) = (\kappa(\chi(\mathbf{X}, 0) - \chi(\mathbf{X}, t)))\left(\frac{1}{1 + e^{\sigma_a(X-X_T)}}\right),\tag{14}$$

where $\kappa$ is a spring constant, and $X_T = 0.001$ is the spatial limit on the horizontal axis of where the tethering force is applied for Equation (9). All reference parameters for this study are listed in Table 1. $E$ was chosen to be of the same magnitude as stiffnesses observed in epithelial and muscular tissue [55], with $\nu$ selected from previous flexible foil studies [56,57]. $T_{max}^{ref}$ is held in proportion to $E$ and tuned to produce undulations that are comparable to observed tail kinematics. Geometric parameters, such as $\lambda$ and $L$, are chosen to be in line with those observed in Reference [3]. The other model parameters were tuned and phenomologically derived to allow for a smooth gradation of muscular activity and to tether the tail.

**Table 1.** Reference model parameters.

| Quantity | Symbol | Reference Value |
|---|---|---|
| Elastic Modulus | $E$ | 10 kPa |
| Poisson ratio | $\eta$ | 0.3 |
| Target spring constant | $\kappa$ | $10^{10}$ Pa |
| Max reference tension magnitude | $T_{max}^{ref}$ | 4000 N |
| Tail length | $L$ | 6.1 cm |
| Wavelength | $\lambda$ | 5 cm |
| Tether transition parameter | $\sigma_a$ | 10,000 |
| Activation transition parameter | $\sigma_b$ | 500 |

### 2.4. Computational Implementation

The computational domain was taken to be $2\,W \times W \times 3\,W$ m$^3$ with periodic boundary conditions, where $W$ is the domain width. The domain was chosen so as to have minimal interaction between the tail and the boundaries of the domain ($W = 0.2$ m). The fixed domain is discretized using adaptive mesh refinement (AMR) with the Immersed Boundary Adaptive Mesh Refinement (IBAMR) open source framework [58]. With IBAMR, the most refined discretization is reserved for portions of the domain where the structure is present and the vorticity magnitude is above a certain threshold. Applying the finest Cartesian grid discretization would result in a $2048 \times 1024 \times 3072$ patch for the entire domain, where the finest spatial grid size is $h = W/1024$. The timestep was $\Delta t = 10^{-4}$. Computational parameters have been reported in Table 2. Benchmark problems for the validation of the IBAMR and IB/FE framework can be found in References [54,58,59].

**Table 2.** Computational parameters.

| Quantity | Symbol | Reference Value |
|---|---|---|
| Numerical timestep | $\Delta t$ | $10^{-4}$ |
| Domain width | $W$ | 0.2 m |
| Grid stepsize | $h$ | $W/1024$ |

## 3. Results

### 3.1. Kinematics and Fluid Flow of Larvacean Tail Beat

Giant larvaceans (genus *Bathochordaeus*) were observed from June to December 2015 using DeepPIV inaging technology (mounted to MiniROV) [3] during cruises aboard RVs *Western Flyer* and *Rachel Carson*. During 13 separate deployments of the instrument, we observed 71 individuals of the genus Bathochordaeus (specifically *B. stygius* and *B. mcnutti*) and collected DeepPIV measurements on 24 individuals, resulting in nearly 49 h of high-definition video. White light illumination (Figure 1A) and DeepPIV measurements (Figure 1B) were collected on a subsample of those individuals. For *B. mcnutti*, 3 observations were selected for subsequent analysis due to the number of tail beat cycles and using the criteria mentioned in the Methods and Materials section (Video S1). Tail morphometrics (e.g., tail width $W$ and length $L$), tail wave characteristics (e.g., amplitude $a$, wavelength $\lambda$,

and frequency $f$), and the number of tail beat cycles (N) were determined for all individuals (Table 3). Refer to Reference [3] for more details on the experimental measurements.

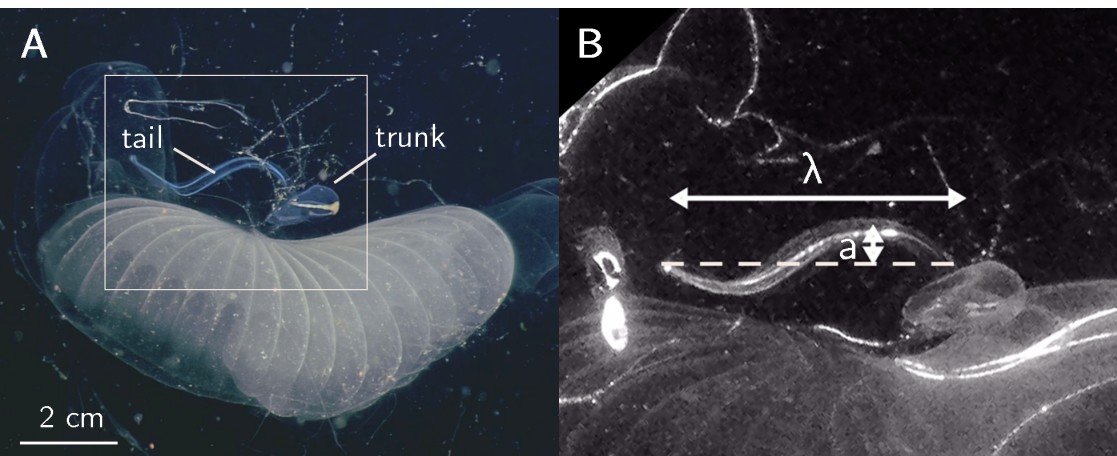

**Figure 1.** In-situ video frame grabs. (**A**) *Bathochordaeus mcnutti* inside its mucus house viewed under white lighting. The white rectangle indicates the area shown in (**B**), which is a close-up view of the animal, illuminated by the laser. Measured tail wave characteristics $\lambda$ (wavelength) and $a$ (amplitude) are shown.

**Table 3.** Kinematics results from in situ observations of *Bathochordaeus mcnutti*. Shown are tail width (W), tail length (L), amplitude (a), wavelength ($\lambda$), frequency (f), and number of consecutive tail beats (N) used for measurement of the kinematics. Modified from Reference [3].

|  | Tail |  |  | Tail Wave |  |  |
|---|---|---|---|---|---|---|
|  | **W** | **L** | **a** | **$\lambda$** | **f** | **N** |
|  | **cm** | **cm** | **cm** | **cm** | **$s^{-1}$** |  |
| BM1 | 3.2 | 6.1 | 1.4 | 5.5 | $0.59 \pm 0.02$ | 3 |
| BM2 | 3.1 | 6.6 | 2.0 | 5.2 | $0.68 \pm 0.05$ | 5 |
| BM3 | 2.7 | 5.8 | 1.7 | 5.2 | $1.04 \pm 0.09$ | 2 |

### 3.2. Model Schematic and Results

In the modeling portion of this study, we examine the role that the activation region of the tail plays on directing fluid flow. This is motivated by the observed kinematics of the larvacean tail (Figure 1, Table 3). Noting that the inflection point on the larvacean tail is positioned at roughly two-thirds of the tail's length (similar to other swimmers and flyers [27]), we can vary the location of this inflection point by varying the activation region $A$ of the tail where a muscle stress (Equation (9)) is applied. The choice of the elastic modulus of the tail (10 kPa) is in the range of what has been observed in the epithelial cells and muscular tissues [55]. These simulations can then be used to explore the effects of differing inflection point locations, i.e., the extent of the muscle activation region, on the resulting wake.

Using the reference parameters of Table 1, we initialize the tethered tail at rest in quiescent flow. A set of six simulations were performed, where we span different extents of the muscle activation region, $A = \frac{1}{6}, \frac{2}{6}, \frac{3}{6}, \frac{4}{6}, \frac{5}{6}$, and $\frac{6}{6}$, with $A_{ref} = \frac{4}{6}$ corresponding to the observed inflection point, hence underlying extent of muscle activation in the larvacean tail. In each of these simulations, the model is driven for 10 cycles of the applied tension waves (Equations (10)–(13)). To ensure a fixed power between the simulations over one cycle of the activation wave, tension magnitude is scaled with respect to the activation region $T_{max} = (A_{ref}/A)T_{max}^{ref}$. The wavelength and driving frequency are chosen to be in line with the observed tail wave (Table 3), with $\lambda = 5$ cm and $f = 1$ Hz. The Reynolds number ($Re = \frac{\rho L U}{\mu}$) for the simulations is 671, using $af$ as the characteristic velocity $U$, where $a$ is the recorded amplitude of the reference case, discussed in the following section.

### 3.2.1. Reference Case

Due to its correspondence with the experimental observation of the inflection point location, we chose the $A_{ref}$ simulation (where the inflection point is located at 2/3 down the tail) as our reference case (see Supplementary Materials Video S2). To better examine the interplay between the passive and active material properties of the tail, in Figure 2, we have plotted the activation strength, $\gamma$, on the tail during the tenth activation cycle. At the start of the cycle (Figure 2a), the tail's deformation is a result of the previous cycle's activation wave, with activation present on both the top and bottom portion of the tail. As the wave of tension travels down the tail (Figure 2b), we note both the spatial extent of activation as a function of $\alpha$, with no activation present in the last third of the tail, due to $A_{ref} = A = \frac{4}{6}$. In the last third of the tail, the resulting kinematics are a product of the passive elastic properties of the tail and the local fluid environment. As the wave travels further down the tail (Figure 2c–e), we note that the active tension wave precedes the changes in the curvature of the tail. This is a result of the interaction between the passive and active material properties, where the elastic material responds to the passage of the active tension wave. At the end of the activation cycle in Figure 2f, the tail's deformation returns to its initial configuration. The resulting kinematics of the cycle correspond to the observed tail wave during pumping, with an amplitude of $a = 1.1$ cm. This amplitude corresponds to the observations of *B. mcnutti* wave forms, though the model amplitude is slightly lower than the observed amplitude due to the tethering force at the tail tip.

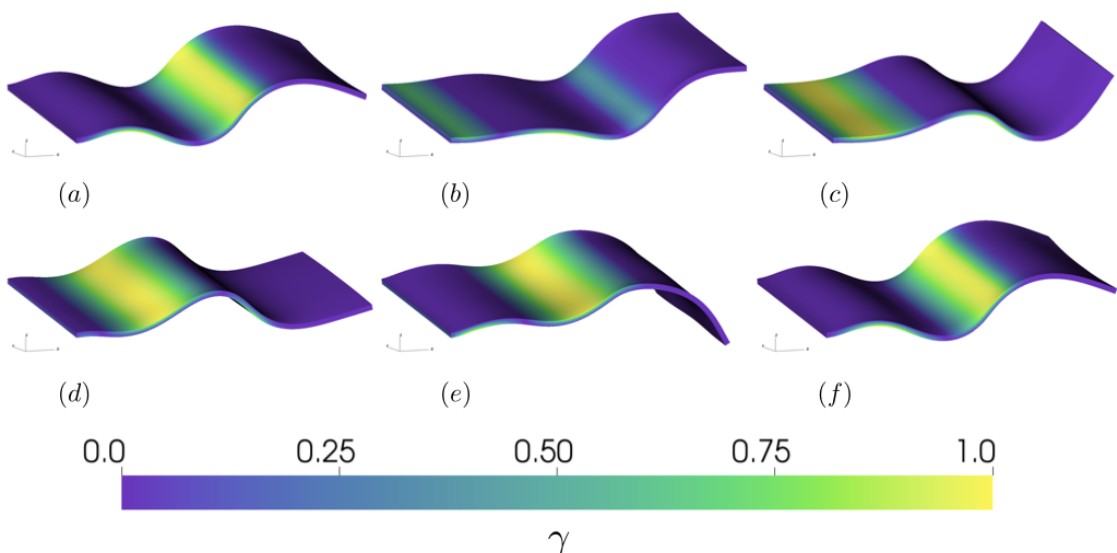

**Figure 2.** Snapshots of the tail during a steady-state activation cycle of the active stress at times (**a**) 9.0, (**b**) 9.2, (**c**) 9.4, (**d**) 9.6, (**e**) 9.8, and (**f**) 10.0. Tail color indicates the instantaneous strength of contraction, $\gamma$, with 1 representing peak applied tension and 0 representing no applied tension. Note that the activation wave is alternating between the two sides of the tail, with the top half fully out of phase with the bottom half.

### 3.2.2. Varying the Activation Region

Varying the extent of the activation region of the tail, $A$, in turn affects the kinematics over the activation cycle. Allowing each simulation to reach a steady-state (after 6 consecutive tail pumping cycles in all cases), we observe differences in the resulting kinematic profile over the activation cycle in Figure 3 (see also Supplementary Materials Video S3). When the activation region is self contained near the base of the tail ($A = \frac{1}{6}, \frac{2}{6}$), the resulting kinematics are strongly determined by the passive elastic properties of the tail and reminiscent of the deflections of pitching and heaving flexible panels [46,56,60,61]. If the activation region extends nearly or completely down the tail ($A = \frac{5}{6}, \frac{6}{6}$), the active stress dominates the motion of the tail and deforms the trailing edge of the tail according to the stress acting on it. For the cases where the activation region extends towards the

midpoint of the tail ($A = \frac{3}{6}, \frac{4}{6}$), the active portion allows for both the tail wave to form as a result of the applied stress and actuates the completely passive trailing edge portion of the tail. Treating the spatial limit of the active portion ($X = AL$) as an inflection point, we measured the relative angle of the inflection point with respect to the trailing edge to be between $-30.7°$ and $32.4°$ for the $A = \frac{4}{6}$ case and between $-25.7°$ and $26.9°$ for the $A = \frac{3}{6}$ case. These angles correspond to the observed inflection angles of other organisms' flexible propulsors [27].

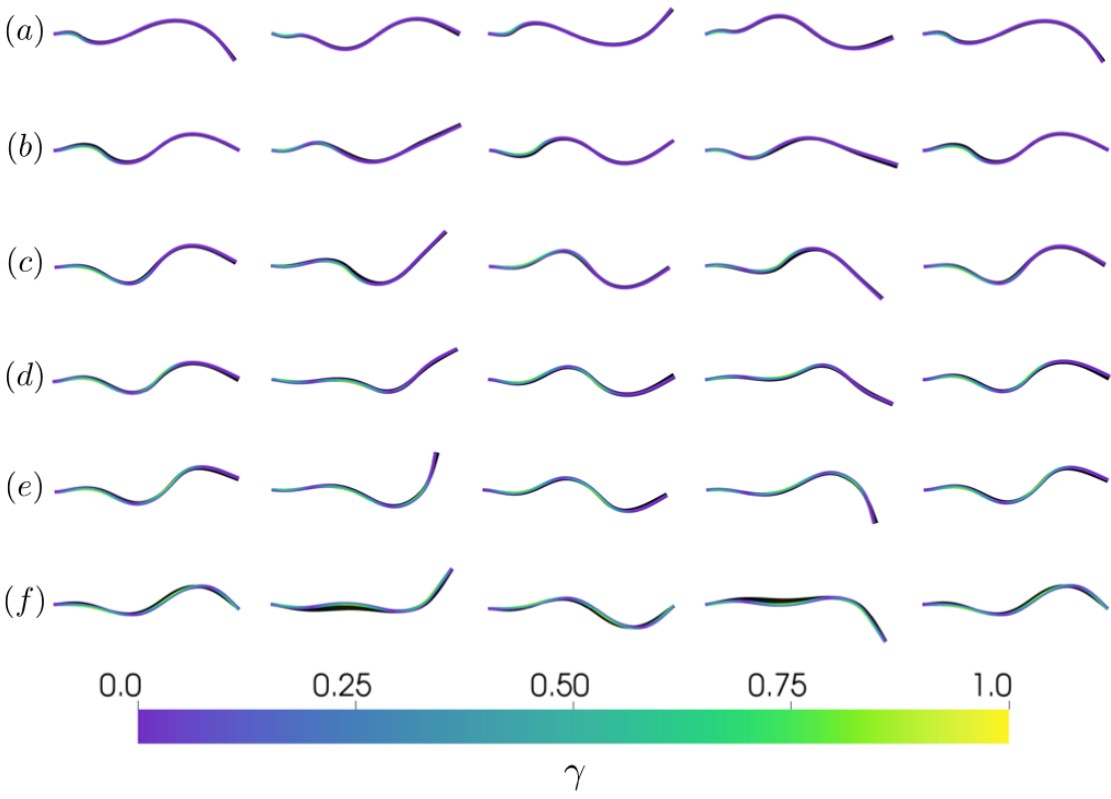

**Figure 3.** Side profile views at time (from left to right) 9.0, 9.25, 9.5, 9.75, and 10.0 s for $A = $ (**a**) $\frac{1}{6}$, (**b**) $\frac{2}{6}$, (**c**) $\frac{3}{6}$, (**d**) $\frac{4}{6}$, (**e**) $\frac{5}{6}$, and (**f**) $\frac{6}{6}$. Tail color indicates the instantaneous strength of contraction ($\gamma$).

To assess the functional impact of the different kinematics, we plotted isocontours of the fluid velocity component $u_x$ in the wake of the tail for differing extents of the activation region (Figure 4), where positive $u_x$ contours correspond to fluid being directed away from the base of the tail. As the tail wave oscillates, fluid is transported towards the tail's trailing edge and is directed into the tail's wake. When the activation region encompasses the majority of the tail ($A = \frac{5}{6}, \frac{6}{6}$; Figure 4e,f), the result of the applied stress deforming the trailing edge yields a wider region of positive fluid velocity (see Supplementary Video S4). Comparing these wakes with the other activation cases, we note that the wakes with lower activation regions have $u_x$ contours that are concentrated along the tail's midline. This suggests that for $A = \frac{5}{6}, \frac{6}{6}$, the transfer of momentum to the downstream-directed flow is reduced thereby lowering the efficiency of fluid transport. Comparing the wakes of the tails with midpoint activation region ($A = \frac{3}{6}, \frac{4}{6}$, Supplementary Video S5) with the tails where the activation region is restricted to the base ($A = \frac{1}{6}, \frac{2}{6}$, see Supplementary Video S6), we find similar wake profiles, although there appears to be a larger region of elevated $u_x$ for $A = \frac{3}{6}, \frac{4}{6}$. Note that the power input for each of the simulations is fixed for all cases.

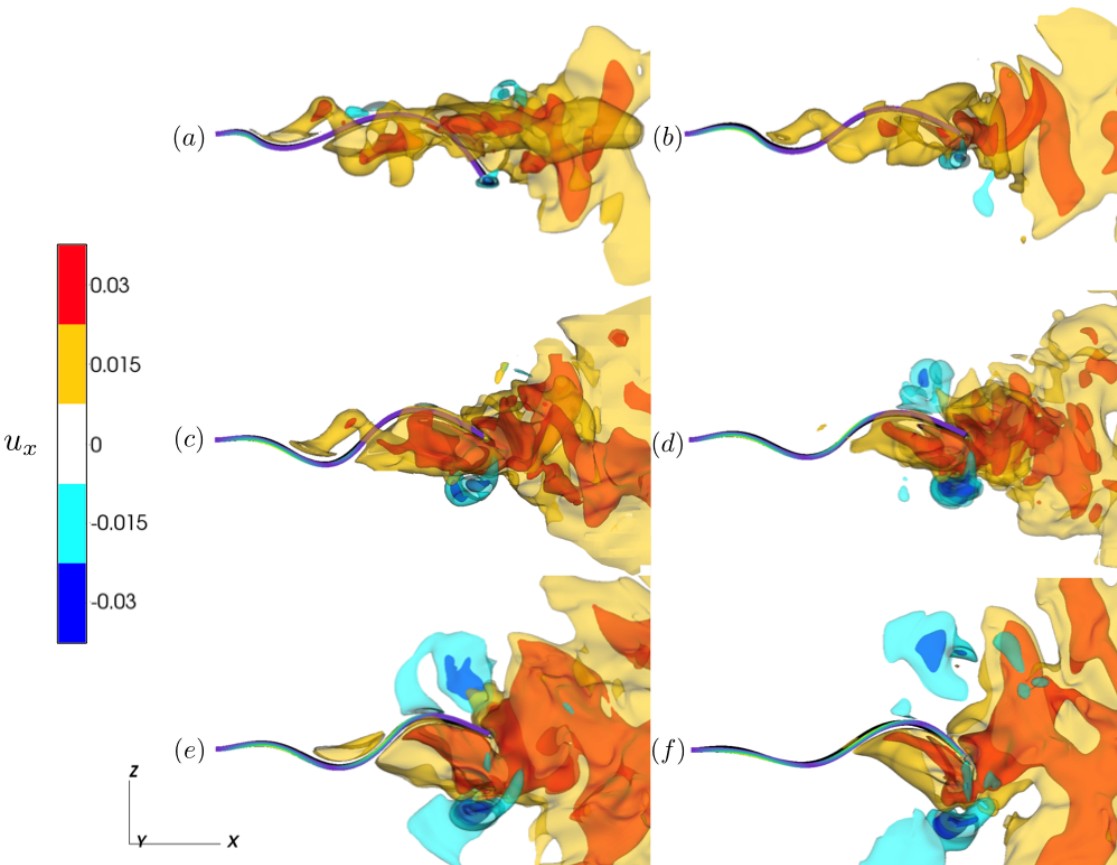

**Figure 4.** Isocontours of $u_x$ at time $t = 10.0$ for $A =$ (**a**) $\frac{1}{6}$, (**b**) $\frac{2}{6}$, (**c**) $\frac{3}{6}$, (**d**) $\frac{4}{6}$, (**e**) $\frac{5}{6}$, and (**f**) $\frac{6}{6}$. Tail color indicates the instantaneous strength of contraction ($\gamma$).

To quantify the role of the activation region's extent on the wake, we examined the average flow speed, $u_x^{avg}$, over a steady state activation cycle in a $2L \times 2L$ region 1 cm behind the trailing edge of the tail at rest. Plotting the average flow speed with respect to the extent of the activation region $A$ in Figure 5, we find that peak flow speeds correspond to $A = \frac{4}{6}$, with elevated flow speeds also present for $A = \frac{3}{6}$ as well. For the largely passive tail ($A = \frac{1}{6}, \frac{2}{6}$), the resulting average flow speeds are significantly less than for tails whose inflection point is near or past the midline. The average flow speed notably decreases as the inflection point moves further down the tail, suggesting an advantage for keeping the inflection point near two-thirds of the lengths of the tail.

Examining the $y$-component of vorticity, $\omega_y$, we note the vortex structures that contribute to fluid transport in the longitudinal direction. Plotting isocontours over a steady-state cycle (Figure 6), we note the alternating vortex structures generated by the trailing edge's undulation over the activation cycle. To further characterize the differences as a result of the activation region, we compared the vortex wake structure of $A = \frac{1}{6}, \frac{4}{6}$, and $\frac{6}{6}$ during the downstroke of the trailing edge at steady state (Figure 7). In all three cases, 2P (2-Pair) wake structures were present [62], with two pairs of vortices shed per activation cycle during the downstroke and upstroke motion of the trailing edge. These wake structure corresponds to 2P wake observed of undulating flexible panels associated with second beam mode resonance Hoover et al. [57], Quinn et al. [61]. Using $A = \frac{4}{6}$ as a reference, we note the increased distance of the shed vortex pairs from the centerline of the tail for $A = \frac{6}{6}$. For $A = \frac{1}{6}$, we note that while the 2P wake structure is still present, the vortex pairs are less defined when compared to $A = \frac{4}{6}$ and $\frac{6}{6}$.

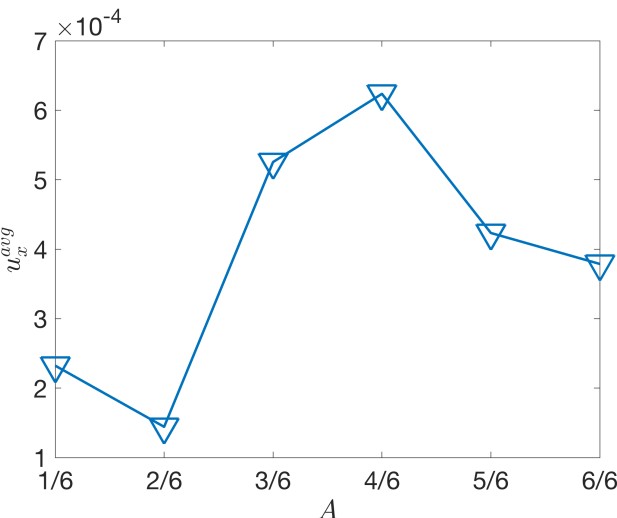

**Figure 5.** Average flow speeds in a 2 $L$ ×2 $L$ region in the wake of the tail plotted with respect to the activation region, $A$.

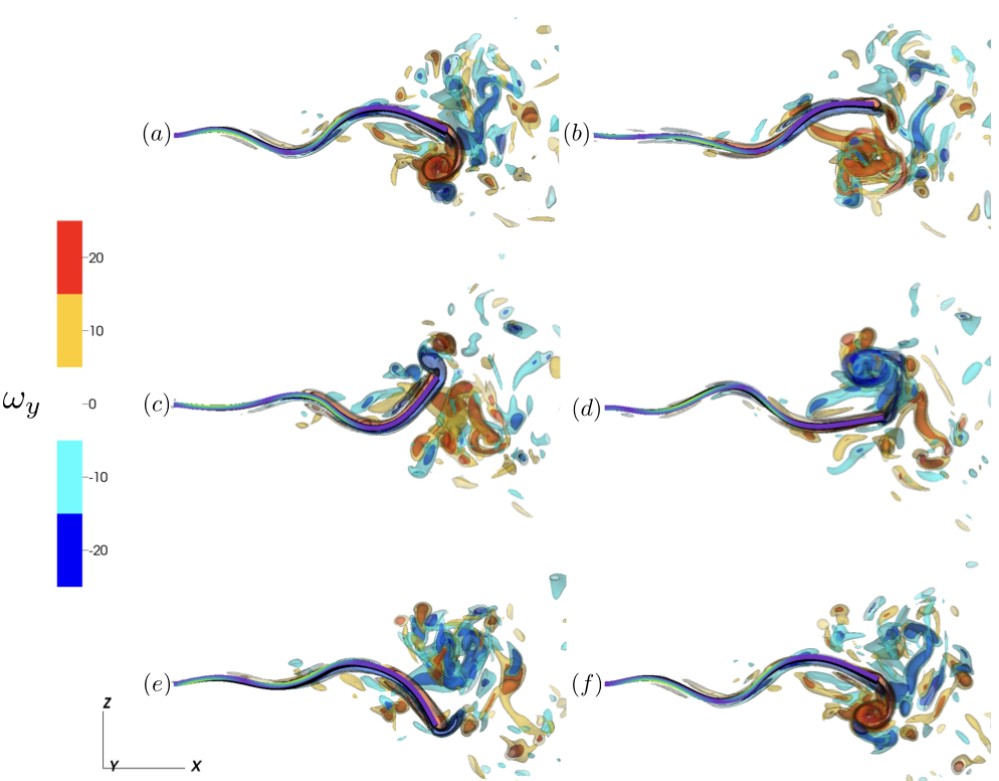

**Figure 6.** Isocontours of the $y$-component of vorticity ($\omega_y$) for $A = \frac{4}{6}$ at $t =$ (**a**) 9.0 s, (**b**) 9.2 s, (**c**) 9.4 s, (**d**) 9.6 s, (**e**) 9.8 s, and (**f**) 10.0 s. For clarity, only the half plane is plotted. Tail color indicates the instantaneous strength of contraction ($\gamma$).

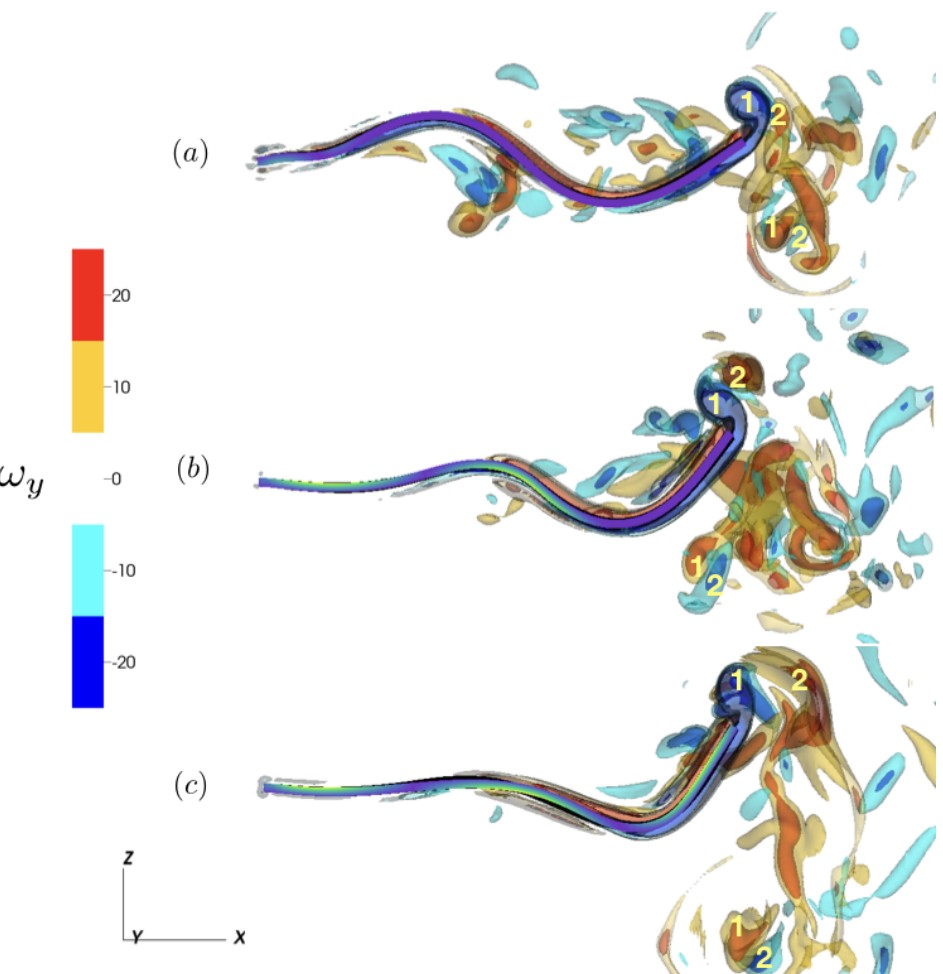

**Figure 7.** Isocontours of the $y$-component of vorticity ($\omega_y$) during the down stroke of the trailing edge for $A =$ (**a**) $\frac{1}{6}$ (t = 9.65 s), (**b**) $\frac{4}{6}$ (t = 9.4 s), and (**c**) $\frac{6}{6}$ (t = 9.35 s). We note the two vortex pairs generated by the tail's trailing edge during a full activation cycle, corresponding to a 2P wake structure. For clarity, only the half plane is plotted. Tail color indicates the instantaneous strength of contraction ($\gamma$).

## 4. Discussion

The computational model evaluated here allows for systematic evaluation of how an inflection point in a propulsor affects fluid transport by comparing the wakes generated by larvacean tails actuated by constant power but variable activation regions. Here, we report that the simulated tail movements of the giant larvacean *Bathochordaeus mcnutti*, where the inflection point is approximately $\frac{2}{3}$ from the base of the tail, results in a downstream-directed wake. This downstream-directed wake is more apparent for active regions of the tail ranging from $\frac{3}{6}$ to $\frac{4}{6}$ (Figure 4) of the total tail length. As our simulations show, this activation pattern then permits the tail tip to passively bend in its interaction with the downstream wake (Figure 3). Furthermore, the resulting average flow speeds for the different activation regions (Figure 5) suggests that having the inflection point at two-thirds of the tail's length results in higher average flow speeds downstream.

Our modeling results show that comparatively simple, dual-layer activation patterns combined with an elastic substrate can create complex and tuneable kinematics and wake structures. In the future, it would be valuable to probe the mechanical properties of the larvacean's tail tip and understand how they modulate the motion of the inflection point during bouts of pumping or feeding. While an inflection point in flapping propulsors is a common feature in biology [27], it remains unknown how the degree of bending depends on passive tissue properties and how this may impact flow transport. Understanding

the flow dynamics resulting from dual-layer muscle activation coupled to flexible tail substrates could inspire novel designs of tissue-engineered soft robots and biomedical functional assays, expanding the current state-of-the-art focus on single-layer muscle actuation [63–65].

Additionally, our numerical model represents one of the first models where the motion of a flexible sheet is the result of internal material properties, rather than an external actuation that moves (e.g., pitches or heaves) the leading edge of the foil [66–69]. By driving the motion of the tail with applied tension, we are able to identify different characteristic behaviors as a function of the activation region. At one extreme ($A = \frac{1}{6}, \frac{2}{6}$), the motion of the tail is dominated by the passive elastic properties of the tail, with the resulting kinematics similar to the deformations of flexible foils with an actuation at the leading edge. In the other extreme ($A = \frac{5}{6}, \frac{6}{6}$), the kinematics are a result of the direct interplay between the active and passive material properties, with the majority of the tail being actuated by an applied stress. In between those regimes is a middle ground ($A = \frac{3}{6}, \frac{4}{6}$), where a tail wave due to the applied stress is present, but the distal, non-activated portion of the tail still exhibits passive deformations resulting from the actuation initiated at the base of the tail.

Noting the similarities of our study to the literature regarding actuated flexible foils, we believe our model to be a strong tool for furthering the biomimetic design principles established in these previous studies. In many of these studies, a flexible hydrofoil or panel takes the place of an animal's flexible appendage and is actuated, either by heaving or pitching its leading edge, in a flow tank [42,60,70]. These studies have broadly examined a number of factors that affect their performance, such as panel geometry [71,72], actuation strategy [69], wall effects [73], and flexibility profile [56,74]. One important finding in Reference [56] was that swimming performance was dependent on the effective flexibility of the actuated panel, a dimensionless value that describes the ratio of added mass forces from the fluid (due to actuation) to internal bending forces. Furthermore, certain effective flexibilities correspond to beam mode resonance in the deflections of the panels [57]. In our study, the cases where the tail is mostly passive ($A = \frac{1}{6}, \frac{2}{6}$) are actuated predominantly at the leading edge of the tail, suggesting that their performance would be dependent on the effective flexibility of the actuated tail. For activation regions that extend further down the tail ($A = \frac{3}{6}, \frac{4}{6}$), the active portion of the tail does not allow for comparison, but the passive trailing portion would still have the potential for further study in the context of the tail's effective flexibility. Additional analysis (e.g., beam-mode [56,57]) on the passive portion of the tail could further our understanding of the broad agreement of propulsor inflection points and angles across taxa [27]. We note that, when the activation region extends through the majority of the tail ($A = \frac{5}{6}, \frac{6}{6}$), thus making it less apt for comparisons to flexible foil studies, the resulting wake of Figure 4 has a wider angle (relative to the centerline of the tail) than the other activation cases.

While the simulations shown here evaluate the fluid transport characteristics of a simplified larvacean model, larvaceans exhibit these kinematics in a constrained environment within their mucus house [3,5]. How the presence of solid boundaries (e.g., the mucus house) impacts or alters the resulting fluid transport is still unknown. The interaction between the wake of the animal and the chamber in which the animal resides is likely to reshape the flow field, and it could be key to understanding the role of the tail's actuation in achieving efficient pumping through the mucus house. Interestingly, the computational models of the larvacean tail suggest that the inflection point may (1) reduce the vertical excursion of the tail tip (Figure 3) and (2) reduce vertical contributions of fluid momentum (Figure 4), thereby minimizing tail-boundary interactions and potentially increasing fluid transport. This is further seen when comparing the 2P wakes of the fully activated tail with one with a defined inflection point (Figure 7), with the two vortex pairs farther away from the center axis for the fully activated tail. An increase in fluid transport could lead to higher filtration rates and an improvement in feeding performance of these animals, and further investigation is required to understand these linkages. It is noteworthy that the larvacean in its mucus house is a rare example of a stationary, dynamic fluid pump in nature [75],

except for ciliated ducts (such as those in sponges) that operate at a lower Reynolds number regime [76]. Future insights learned from giant larvaceans may improve our understanding of structure-function relationships in biological dynamic pumps operating in a constrained space, as well as may even inspire the engineering of biologically inspired actuators for moving fluid through conduits.

Finally, we would like to emphasize the value of this work in demonstrating a novel workflow for non-invasively studying living, deep sea animals that are usually difficult to access and maintain in the laboratory. In this study, we utilized imaging and modeling methodologies to investigate the fluid transport function of giant larvacean specimens from the deep, dark waters of the mesopelagic zone. These technological advances can now also be applied to investigate other rarely studied swimmers, such as ecologically important and diverse siphonophores and other gelata [77], and grow our limited knowledge of these and other "hidden" (i.e., non-model) organisms [78]. Beyond providing new insights into the lives of individual species, our approach can also be tied into the greater effort of achieving a better understanding of the mesopelagic ecosystem, which is one of the most understudied habitats that is vital to the ocean's health and productivity [79]. Recent studies suggest that the mesopelagic zone is particularly vulnerable to accelerated ocean warming, yet little to nothing is known about how this ecosystem might respond to rising temperatures [80]. Integrated research workflows involving biologists, physicists, and engineers, such as what was presented in this work, will be crucial to addressing these and other pressing challenges in ocean science.

**Supplementary Materials:** The following are available online at https://www.mdpi.com/2311-5521/6/2/88/s1, Video S1: White-illuminated (color) and laser-illuminated (monochrome) DeepPIV footage of *Bathochordaeus mcnutti*. Video S2: Video of the tail deformations for the $A_{ref}$ case for 10 activation cycles, with the color indicating the instantaneous strength of contraction plotted on the tail. Video S3: Video of the tail deformations from side profile view for (top to bottom) $A = \frac{1}{6}, \frac{2}{6}, \frac{3}{6}, \frac{4}{6}, \frac{5}{6}, \frac{6}{6}$ for 10 activation cycles, with the color indicating the instantaneous strength of contraction plotted on the tail. Video S4: Video of the isocontours of $u_x$ during the first 10 activation cycles for the reference case $A = A_{ref} = \frac{4}{6}$. Video S5: Video of the isocontours of $u_x$ during the first 10 activation cycles for the reference case $A = \frac{1}{6}$. Video S6: Video of the isocontours of $u_x$ during the first 10 activation cycles for the reference case $A = \frac{6}{6}$.

**Author Contributions:** Conceptualization, methodology, formal analysis, investigation, writing, and original draft preparation, A.P.H., J.C.N., and K.K.; visualization, A.P.H., J.C.N., and J.D.; software, A.P.H.; data curation, validation, writing, review and editing, all authors; resources, supervision, project administration, funding acquisition, K.K. All authors have read and agreed to the published version of the manuscript.

**Funding:** This research was funded by the David and Lucile Packard Foundation.

**Institutional Review Board Statement:** Not applicable.

**Informed Consent Statement:** Not applicable.

**Data Availability Statement:** The experimental data that inspired this study is available in literature [3].

**Acknowledgments:** The authors are thankful for the contributions made by D. Graves, C. Kecy, D. Klimov, J. Erickson, A. Sherman, and technical staff, as well as the crews of RVs *Rachel Carson* and *Western Flyer*, and the pilots of ROVs *Doc Ricketts*, *Ventana*, and *MiniROV*.

**Conflicts of Interest:** The authors declare no conflict of interest.

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
