# Peer review of "A Computational Model for Tail Undulation and Fluid Transport in the Giant Larvacean"

_fluids, doi:10.3390/fluids6020088_

Round 1

Reviewer 1 Report

Comments

Papers like this are a pleasure to read: an insightful synthesis of the existing literature, a clear motivation for the study, an elegant approach involving state-of-the-art in situ experiments and well-benchmarked numerics, compelling results, and a forward-looking Discussion with clear and concise communication throughout.  

Synopsis

The authors use in situ high-speed video and PIV observations to measure geometry, tail kinematics, and filtration rates in giant larvaceans. These measurements are used to parameterize an elastohydrodynamic module, in which the tail kinematics emerge as a function of the local hydrodynamics and elastic stresses, both passive stresses due to straining of the tail surface and active stresses due to an imposed wave of “muscle” tension propagating from the tethered base of the tail. The spatial extent of the applied active stress (activation region) is varied among model runs and its effects on the strength and spatial extent of the wake structure are quantified. The results show optimization for an activation region around 2/3, corresponding well with both measurements of giant larvacean kinematics that show a distinct inflection point at this location (2/3 the length of the tail from the base) and the seemingly universal 2/3 inflection point seen in numerous flapping swimmers and flyers. Unique contributions include not only in situ behavioral, morphological, and PIV measurements on delicate deep sea giant larvaceans, but also an investigation of the couple hydrodynamics and biomechanics of an (rare) organism whose flapping appendage drives both propulsion and feeding. Additionally, thoughtful syntheses of the existing literature and future directions add quite a lot of value.

Introduction

  • Line 28: refs aren’t dynamically linked from the text to the References section.
  • Lines 40-52: refs aren’t dynamically linked from the text to the References section.
  • Line 67: what are those Reynolds numbers (ballpark, or specifically based on the current work/previous works by one of the authors)? Strouhal numbers?
  • Lines 67-71: Some elaboration here would add value. For example, are the differing tail kinematics across body sizes related to Re constraints, differing musculature, etc.? Similarly, for the (implied) lack of the 2/3 inflection point for smaller larvaceans?
  • Lines 84-86: Sentence is too long and doesn’t read very clearly, suggest breaking into 2 sentences for clarity. Something like… By using the giant larvacean mcnutti as a unique model system, we can understand the interactions between the tail’s structural rigidity and flexibility, muscle actuation, and fluid forces that underlie tail kinematics, including the pronounced 2/3 inflection point. This will shed light on how the inflection point mechanism, including the ubiquitous 2/3 inflection point seen in numerous flapping propulsors, impacts fluid transport & feeding performance in an animal that is both swimming & feeding.”

Materials & Methods

  • Lines 102, 103, and 106 : missing an “x” in the area dimensions
  • Line 110: maybe a quick elaboration on the means of evaluating whether the laser sheet was truly bisecting the organism along a line of symmetry, given the authors previous point about the importance of symmetry-breaking w/r/t assumptions of two-dimensionality in these types flows (e.g. Line 42)?
  • Paragraph beginning on Line 115: I realize some of these details are likely covered in the original DeepPIV paper (Ref 28, that the authors do reference throughout this paragraph), but it might be useful to briefly elaborate a bit more on how the fluid velocities were computed. E.g. Is the local velocity computed as the particle streak length in a single frame divided by the image exposure time? Or is the particle streak path integrated over multiple frames, and if so how does this (maximum) integration timescale compare with say the pumping period of the tail or some other relevant transient timescale? Similarly, some additional elaboration on the mapping from local flow velocities to volumetric flow rates would be valuable. Are the (sparse) local velocities as derived from the particle strength lengths spatially interpolated and then integrated over an estimated cross-sectional flow area (of the tail chamber)? This seems like an important point of clarity, given the need to estimate an out-of-plane (I think, based orientation of the images in Fig. 1) morphological dimension to derive the corresponding flow area for the volumetric filtration rate computation.
  • Line 122 (also Table 3): I find the use of “flow rate” to describe fluid velocities really confusing; maybe it’s just me but “flow rate” connotes volumetric flow. Applies throughout the manuscript.
  • Table 1: With the exception of the geometric parameters, are the other model parameters physiologically realistic or are their values primarily the result of turning knobs to get the model to reproduce observed tail kinematics?

Results

  • Line 186 & Figure 1: Given the importance of filtration/pumping rates, I think the tail chamber geometry (location/extent + dimensions) could be better illustrated. I assume the tail chamber height H is in the image plane and the width W is out-of-plane? I’m curious how the out-of-plane dimension was estimated (see previous comment about paragraph starting on line 115). Maybe some additional dimensioning etc. in the right panel of Fig.1 or even a third panel focusing the tail chamber itself, rather than the tail itself?
  • End of Line 212: This sentence isn’t super clear to me. Also, how are alpha (spatial extent of applied tension in transverse direction) and activation region A related/different?
  • 4: It would be really interesting to extend the domain of the model to the region upstream of the “tail” to see where, with the additional constraint of the body/tail chamber boundary conditions, the fluid that the larvacean captures/filters comes from. Obviously beyond the scope here, just a general comment.

Discussion

  • In general, I really liked the tone taken in the Discussion: a mix of contextualizing the current study in the literature with a good emphasis on the novel contributions, and considerable discussion towards new questions motivated and new doors opened with the toolbox developed in the paper.
  • One thing I found lacking in the Discussion (in general) was a more mechanistic explanation of the hydrodynamics of the wake structure, given the larger context of flapping foils and flapping biological propulsors that the authors establish throughout. For example, what are some common features of the tip vortices shed from the trailing edge of the “tail”? How do the kinematics of the trailing edge tip influence tip vortex shedding and the resulting wake structure/propulsive efficiency? How might the inflection point/passive trailing end tail segment enhance/alter tip vortex shedding and/or the formation of (or lack thereof) a coherent wake jet? I realize the authors briefly mention some of these topics in the Results/Discussion sections (e.g. Lines 242-252, Line 319), but I feel a deeper dive into the diverse hydrodynamic phenomena at play (perhaps formation/shedding of tip vortices, vortex instability/breakdown, emergence of a coherent wake jet, etc.?) would add value here. I realize there is a vast literature on this subject, and as the authors are experts and contributors to this body of literature, their insights and contextualization of the hydrodynamic phenomena at play here would be much appreciated!
  • Bunch of the dynamic links to figs. on pg. 9 are broken.

Reviewer 2 Report

Please find below my review of the paper, `` Bending modes for effective fluid transport: The case of the giant larvacean’’ for consideration for publication as a research article in Fluids. In this paper, the authors use the immersed boundary method to numerically simulate the motion and resulting hydrodynamics of the beating tail of a giant larvacean. A deep-sea robot and imaging tools were also used to obtain and validate the resulting kinematics. One of the unique features of the study is that the tail’s kinematics are not prescribed, they emerge from the muscle activation model coupled with the elastic property of the tail and the hydrodynamic forces.

Overall, the paper presents a state-of-the-art study that combines robotics and numerical simulations to reveal the fluid dynamics of an organism that is not at all well understood. I believe that the paper should be of interest to the reader of fluids, particularly those who work in the areas of biomechanics, biomimetics, and marine biology. I have a couple of comments that should be addressed that mostly focus on better articulating what is and is not known about larvaceans and how parameters were chosen without that information.

Major comments

  • Introduction - More details should be provided regarding what is known about the musculature of the larvacean tail, its activation pattern, and the material properties of the tail. If no or limited information is available, that should be explained too.
  • Line 68 – “…reveal differing tail kinematics during in-house pumping” – please describe these differences in more detail. Also, are these differences between in-house beating and swimming? Or are there differences between species? Or both?
  • Line 126 – “we assume that the tail wave is driven by a wave of active tension traveling from the base of the tail to the inflection ” – can you explain the justification for this in a bit more detail (e.g. this is based on known physiology or this was mathematically determined to generate the correct kinematics)?
  • Line 152 – “assumed material properties of the tail” – Can you say a bit more about this? Likely, no one has measured the material properties of larvacean tails. However, there are probably some estimates that can be done for its width and an approximate elastic modulus based on other gelatinous organisms.
  • Line 154 – “the opposing action of two parallel muscle layers” --- are these known to exist?
  • Please discuss in a bit more detail how T_max chosen to generate the appropriate amount of tail deformation (and how this depends on the choice of the elastic modulus).
  • Please report the Re range of these simulations since the particular method used is best at lower Re. How does this compare to actual larvaceans?

Minor comments

Line 19  – “For this reason, there is significant interest in studies of biological fluid transport to leverage nature’s R&D for robotics and bioinspired design” – needs a reference.

Line 22 – “Recent developments that improve access to other biological” --- this also needs a ref.

Line 29 – “implications for development of engineered systems” -> “implications for the development of engineered systems..”

Line 38 - “reduction” -> “the reduction”

Line 41 – “understanding of these systems” -> “understanding these systems”

Line 67 – “thereby spanning a range of Reynolds numbers.” --- provide that range of Re here.

Line 213 – “the the spatial extent” -> the spatial extent

Line 306 – “portion would be still have” -> “portion would still have”
